# Mixed Optimization for Smooth Functions

**Mehrdad Mahdavi**      **Lijun Zhang**      **Rong Jin**

Department of Computer Science and Engineering, Michigan State University, MI, USA

{mahdavim,zhanglij,rongjin}@msu.edu

## Abstract

It is well known that the optimal convergence rate for stochastic optimization of smooth functions is $O(1/\sqrt{T})$, which is same as stochastic optimization of Lipschitz continuous convex functions. This is in contrast to optimizing smooth functions using full gradients, which yields a convergence rate of $O(1/T^2)$. In this work, we consider a new setup for optimizing smooth functions, termed as **Mixed Optimization**, which allows to access both a stochastic oracle and a full gradient oracle. Our goal is to significantly improve the convergence rate of stochastic optimization of smooth functions by having an additional small number of accesses to the full gradient oracle. We show that, with an $O(\ln T)$ calls to the full gradient oracle and an $O(T)$ calls to the stochastic oracle, the proposed mixed optimization algorithm is able to achieve an optimization error of $O(1/T)$.

## 1 Introduction

Many machine learning algorithms follow the framework of empirical risk minimization, which often can be cast into the following generic optimization problem

$$\min_{\mathbf{w}\in\mathcal{W}} \ \mathcal{G}(\mathbf{w}) := \frac{1}{n}\sum_{i=1}^{n} g_i(\mathbf{w}), \tag{1}$$

where $n$ is the number of training examples, $g_i(\mathbf{w})$ encodes the loss function related to the $i$th training example $(\mathbf{x}_i, y_i)$, and $\mathcal{W}$ is a bounded convex domain that is introduced to regularize the solution $\mathbf{w} \in \mathcal{W}$ (i.e., the smaller the size of $\mathcal{W}$, the stronger the regularization is). In this study, we focus on the learning problems for which the loss function $g_i(\mathbf{w})$ is smooth. Examples of smooth loss functions include least square with $g_i(\mathbf{w}) = (y_i - \langle \mathbf{w}, \mathbf{x}_i \rangle)^2$ and logistic regression with $g_i(\mathbf{w}) = \log\left(1 + \exp(-y_i\langle \mathbf{w}, \mathbf{x}_i \rangle)\right)$. Since the regularization is enforced through the restricted domain $\mathcal{W}$, we did not introduce a $\ell_2$ regularizer $\lambda\|\mathbf{w}\|^2/2$ into the optimization problem and as a result, we do not assume the loss function to be strongly convex. We note that a small $\ell_2$ regularizer does NOT improve the convergence rate of stochastic optimization. More specifically, the convergence rate for stochastically optimizing a $\ell_2$ regularized loss function remains as $O(1/\sqrt{T})$ when $\lambda = O(1/\sqrt{T})$ [11, Theorem 1], a scenario that is often encountered in real-world applications.

A preliminary approach for solving the optimization problem in (1) is the batch gradient descent (GD) algorithm [16]. It starts with some initial point, and iteratively updates the solution using the equation $\mathbf{w}_{t+1} = \Pi_{\mathcal{W}}(\mathbf{w}_t - \eta\nabla\mathcal{G}(\mathbf{w}_t))$, where $\Pi_{\mathcal{W}}(\cdot)$ is the orthogonal projection onto the convex domain $\mathcal{W}$. It has been shown that for smooth objective functions, the convergence rate of standard GD is $O(1/T)$ [16], and can be improved to $O(1/T^2)$ by an accelerated GD algorithm [15, 16, 18]. The main shortcoming of GD method is its high cost in computing the full gradient $\nabla\mathcal{G}(\mathbf{w}_t)$ when the number of training examples is large. Stochastic gradient descent (SGD) [3, 13, 21] alleviates this limitation of GD by sampling one (or a small set of) examples and computing a stochastic (sub)gradient at each iteration based on the sampled examples. Since the computational cost of SGD per iteration is independent of the size of the data (i.e., $n$), it is usually appealing for large-scale learning and optimization.

While SGD enjoys a high computational efficiency per iteration, it suffers from a slow convergence rate for optimizing smooth functions. It has been shown in [14] that the effect of the stochastic noise

| Setting | Full (GD) | | | Stochastic (SGD) | | | Mixed Optimization | | |
|---|---|---|---|---|---|---|---|---|---|
| | Convergence | $\mathcal{O}_s$ | $\mathcal{O}_f$ | Convergence | $\mathcal{O}_s$ | $\mathcal{O}_f$ | Convergence | $\mathcal{O}_s$ | $\mathcal{O}_f$ |
| Lipschitz | $\frac{1}{\sqrt{T}}$ [1] | 0 | $T$ | $\frac{1}{\sqrt{T}}$ | $T$ | 0 | — | — | — |
| Smooth | $\frac{1}{T^2}$ | 0 | $T$ | $\frac{1}{\sqrt{T}}$ | $T$ | 0 | $\frac{1}{T}$ | $T$ | $\log T$ |

Table 1: The convergence rate ($O$), number of calls to stochastic oracle ($\mathcal{O}_s$), and number of calls to full gradient oracle ($\mathcal{O}_f$) for optimizing Lipschitz continuous and smooth convex functions, using full GD, SGD, and mixed optimization methods, measured in the number of iterations $T$.

cannot be decreased with a better rate than $O(1/\sqrt{T})$ which is significantly worse than GD that uses the full gradients for updating the solutions and this limitation is also valid when the target function is smooth. In addition, as we can see from Table 1, for general Lipschitz functions, SGD exhibits the same convergence rate as that for the smooth functions, implying that smoothness is essentially not very useful and can not be exploited in stochastic optimization. The slow convergence rate for stochastically optimizing smooth loss functions is mostly due to the variance in stochastic gradients: unlike the full gradient case where the norm of a gradient approaches to zero when the solution is approaching to the optimal solution, in stochastic optimization, the norm of a stochastic gradient is constant even when the solution is close to the optimal solution. It is the variance in stochastic gradients that makes the convergence rate $O(1/\sqrt{T})$ unimprovable in smooth setting [14, 1].

In this study, we are interested in designing an efficient algorithm that is in the same spirit of SGD but can effectively leverage the smoothness of the loss function to achieve a significantly faster convergence rate. To this end, we consider a new setup for optimization that allows us to interplay between stochastic and deterministic gradient descent methods. In particular, we assume that the optimization algorithm has an access to two oracles:

- A stochastic oracle $\mathcal{O}_s$ that returns the loss function $g_i(\mathbf{w})$ and its gradient based on the sampled training example $(\mathbf{x}_i, y_i)$ [2], and
- A full gradient oracle $\mathcal{O}_f$ that returns the gradient $\nabla \mathcal{G}(\mathbf{w})$ for any given solution $\mathbf{w} \in \mathcal{W}$.

We refer to this new setting as *mixed optimization* in order to distinguish it from both stochastic and full gradient optimization models. The key question we examined in this study is:

> *Is it possible to improve the convergence rate for stochastic optimization of smooth functions by having a small number of calls to the full gradient oracle $\mathcal{O}_f$?*

We give an affirmative answer to this question. We show that with an additional $O(\ln T)$ accesses to the full gradient oracle $\mathcal{O}_f$, the proposed algorithm, referred to as MIXEDGRAD, can improve the convergence rate for stochastic optimization of smooth functions to $O(1/T)$, the same rate for stochastically optimizing a strongly convex function [11, 19, 23]. MIXEDGRAD builds off on multi-stage methods [11] and operates in epochs, but involves novel ingredients so as to obtain an $O(1/T)$ rate for smooth losses. In particular, we form a sequence of strongly convex objective functions to be optimized at each epoch and decrease the amount of regularization and shrink the domain as the algorithm proceeds. The full gradient oracle $\mathcal{O}_f$ is only called at the beginning of each epoch.

Finally, we would like to distinguish mixed optimization from hybrid methods that use growing sample-sizes as optimization method proceeds to gradually transform the iterates into the full gradient method [9] and batch gradient with varying sample sizes [6], which unfortunately make the iterations to be dependent to the sample size $n$ as opposed to SGD. In contrast, MIXEDGRAD is as an alternation of deterministic and stochastic gradient steps, with different of frequencies for each type of steps. Our result for mixed optimization is useful for the scenario when the full gradient of the objective function can be computed relatively efficient although it is still significantly more expensive than computing a stochastic gradient. An example of such a scenario is distributed computing where the computation of full gradients can be speeded up by having it run in parallel on many machines with each machine containing a relatively small subset of the entire training data. Of course, the latency due to the communication between machines will result in an additional cost for computing the full gradient in a distributed fashion.

**Outline** The rest of this paper is organized as follows. We begin in Section 2 by briefly reviewing the literature on deterministic and stochastic optimization. In Section 3, we introduce the necessary definitions and discuss the assumptions that underlie our analysis. Section 4 describes the MIXED-GRAD algorithm and states the main result on its convergence rate. The proof of main result is given in Section 5. Finally, Section 6 concludes the paper and discusses few open questions.

## 2 More Related Work

**Deterministic Smooth Optimization** The convergence rate of gradient based methods usually depends on the analytical properties of the objective function to be optimized. When the objective function is strongly convex and smooth, it is well known that a simple GD method can achieve a linear convergence rate [5]. For a non-smooth Lipschitz-continuous function, the optimal rate for the first order method is only $O(1/\sqrt{T})$ [16]. Although $O(1/\sqrt{T})$ rate is not improvable in general, several recent studies are able to improve this rate to $O(1/T)$ by exploiting the special structure of the objective function [18, 17]. In the full gradient based convex optimization, smoothness is a highly desirable property. It has been shown that a simple GD achieves a convergence rate of $O(1/T)$ when the objective function is smooth, which is further can be improved to $O(1/T^2)$ by using the accelerated gradient methods [15, 18, 16].

**Stochastic Smooth Optimization** Unlike the optimization methods based on full gradients, the smoothness assumption was not exploited by most stochastic optimization methods. In fact, it was shown in [14] that the $O(1/\sqrt{T})$ convergence rate for stochastic optimization cannot be improved even when the objective function is smooth. This classical result is further confirmed by the recent studies of composite bounds for the first order optimization methods [2, 12]. The smoothness of the objective function is exploited extensively in mini-batch stochastic optimization [7, 8], where the goal is not to improve the convergence rate but to reduce the variance in stochastic gradients and consequentially the number of times for updating the solutions [24]. We finally note that the smoothness assumption coupled with the strong convexity of function is beneficial in stochastic setting and yields a geometric convergence in *expectation* using Stochastic Average Gradient (SAG) and Stochastic Dual Coordinate Ascent (SDCA) algorithms proposed in [20] and [22], respectively.

## 3 Preliminaries

We use bold-face letters to denote vectors. For any two vectors $\mathbf{w}, \mathbf{w}' \in \mathcal{W}$, we denote by $\langle \mathbf{w}, \mathbf{w}' \rangle$ the inner product between $\mathbf{w}$ and $\mathbf{w}'$. Throughout this paper, we only consider the $\ell_2$-norm. We assume the objective function $\mathcal{G}(\mathbf{w})$ defined in (1) to be the average of $n$ convex loss functions. The same assumption was made in [20, 22]. We assume that $\mathcal{G}(\mathbf{w})$ is minimized at some $\mathbf{w}_* \in \mathcal{W}$. Without loss of generality, we assume that $\mathcal{W} \subset \mathbb{B}_R$, a ball of radius $R$. Besides convexity of individual functions, we will also assume that each $g_i(\mathbf{w})$ is $\beta$-*smooth* as formally defined below [16].

**Definition 1** (Smoothness). *A differentiable loss function $f(\mathbf{w})$ is said to be $\beta$-smooth with respect to a norm $\|\cdot\|$, if it holds that*

$$f(\mathbf{w}) \leq f(\mathbf{w}') + \langle \nabla f(\mathbf{w}'), \mathbf{w} - \mathbf{w}' \rangle + \frac{\beta}{2} \|\mathbf{w} - \mathbf{w}'\|^2, \quad \forall \, \mathbf{w}, \mathbf{w}' \in \mathcal{W},$$

The smoothness assumption also implies that $\langle \nabla f(\mathbf{w}) - \nabla f(\mathbf{w}'), \mathbf{w} - \mathbf{w}' \rangle \leq \beta \|\mathbf{w} - \mathbf{w}'\|^2$ which is equivalent to $\nabla f(\mathbf{w})$ being $\beta$-Lipschitz continuous.

In stochastic first-order optimization setting, instead of having direct access to $\mathcal{G}(\mathbf{w})$, we only have access to a stochastic gradient oracle, which given a solution $\mathbf{w} \in \mathcal{W}$, returns the gradient $\nabla g_i(\mathbf{w})$ where $i$ is sampled uniformly at random from $\{1, 2, \cdots, n\}$. The goal of stochastic optimization to use a bounded number $T$ of oracle calls, and compute some $\bar{\mathbf{w}} \in \mathcal{W}$ such that the optimization error, $\mathcal{G}(\bar{\mathbf{w}}) - \mathcal{G}(\mathbf{w}^*)$, is as small as possible.

In the mixed optimization model considered in this study, we first relax the stochastic oracle $\mathcal{O}_s$ by assuming that it will return a randomly sampled loss function $g_i(\mathbf{w})$, instead of the gradient $\nabla g_i(\mathbf{w})$ for a given solution $\mathbf{w}$ [3]. Second, we assume that the learner also has an access to the full gradient oracle $\mathcal{O}_f$. Our goal is to significantly improve the convergence rate of stochastic gradient descent (SGD) by making a small number of calls to the full gradient oracle $\mathcal{O}_f$. In particular, we show that by having only $O(\log T)$ accesses to the full gradient oracle and $O(T)$ accesses to the stochastic oracle, we can tolerate the noise in stochastic gradients and attain an $O(1/T)$ convergence rate for optimizing smooth functions.

**Algorithm 1** MIXEDGRAD

**Input:** step size $\eta_1$, domain size $\Delta_1$, the number of iterations $T_1$ for the first epoch, the number of epoches $m$, regularization parameter $\lambda_1$, and shrinking parameter $\gamma > 1$

1: Initialize $\bar{\mathbf{w}}_1 = \mathbf{0}$
2: **for** $k = 1, \ldots, m$ **do**
3:     Construct the domain $\mathcal{W}_k = \{\mathbf{w} : \mathbf{w} + \mathbf{w}_k \in \mathcal{W}, \|\mathbf{w}\| \le \Delta_k\}$
4:     Call the full gradient oracle $\mathcal{O}_f$ for $\nabla \mathcal{G}(\bar{\mathbf{w}}_k)$
5:     Compute $\mathbf{g}_k = \lambda_k \bar{\mathbf{w}}_k + \nabla \mathcal{G}(\bar{\mathbf{w}}_k) = \lambda_k \bar{\mathbf{w}}_k + \frac{1}{n} \sum_{i=1}^{n} \nabla g_i(\bar{\mathbf{w}}_k)$
6:     Initialize $\mathbf{w}_k^1 = \mathbf{0}$
7:     **for** $t = 1, \ldots, T_k$ **do**
8:         Call stochastic oracle $\mathcal{O}_s$ to return a randomly selected loss function $g_{i_k^t}(\mathbf{w})$
9:         Compute the stochastic gradient as $\hat{\mathbf{g}}_k^t = \mathbf{g}_k + \nabla g_{i_k^t}(\mathbf{w}_k^t + \bar{\mathbf{w}}_k) - \nabla g_{i_k^t}(\bar{\mathbf{w}}_k)$
10:        Update the solution by

$$\mathbf{w}_k^{t+1} = \arg\max_{\mathbf{w} \in \mathcal{W}_k} \eta_k \langle \mathbf{w} - \mathbf{w}_k^t, \hat{\mathbf{g}}_k^t + \lambda_k \mathbf{w}_k^t \rangle + \frac{1}{2}\|\mathbf{w} - \mathbf{w}_k^t\|^2$$

11:     **end for**
12:     Set $\widetilde{\mathbf{w}}_{k+1} = \frac{1}{T_k+1} \sum_{t=1}^{T_k+1} \mathbf{w}_k^t$ and $\bar{\mathbf{w}}_{k+1} = \bar{\mathbf{w}}_k + \widetilde{\mathbf{w}}_{k+1}$
13:     Set $\Delta_{k+1} = \Delta_k/\gamma$, $\lambda_{k+1} = \lambda_k/\gamma$, $\eta_{k+1} = \eta_k/\gamma$, and $T_{k+1} = \gamma^2 T_k$
14: **end for**
**Return** $\bar{\mathbf{w}}_{m+1}$

The analysis of the proposed algorithm relies on the strong convexity of intermediate loss functions introduced to facilitate the optimization as given below.

**Definition 2** (Strong convexity). *A function $f(\mathbf{w})$ is said to be $\alpha$-strongly convex w.r.t a norm $\|\cdot\|$, if there exists a constant $\alpha > 0$ (often called the modulus of strong convexity) such that it holds*

$$f(\mathbf{w}) \ge f(\mathbf{w}') + \langle \nabla f(\mathbf{w}'), \mathbf{w} - \mathbf{w}' \rangle + \frac{\alpha}{2}\|\mathbf{w} - \mathbf{w}'\|^2, \quad \forall\, \mathbf{w}, \mathbf{w}' \in \mathcal{W}$$

## 4 Mixed Stochastic/Deterministic Gradient Descent

We now turn to describe the proposed mixed optimization algorithm and state its convergence rate. The detailed steps of MIXEDGRAD algorithm are shown in Algorithm 1. It follows the epoch gradient descent algorithm proposed in [11] for stochastically minimizing strongly convex functions and divides the optimization process into $m$ epochs, but involves novel ingredients so as to obtain an $O(1/T)$ convergence rate. The key idea is to introduce a $\ell_2$ regularizer into the objective function to make it strongly convex, and gradually reduce the amount of regularization over the epochs. We also shrink the domain as the algorithm proceeds. We note that reducing the amount of regularization over time is closely-related to the classic proximal-point algorithms. Throughout the paper, we will use the subscript for the index of each epoch, and the superscript for the index of iterations within each epoch. Below, we describe the key idea behind MIXEDGRAD.

Let $\bar{\mathbf{w}}_k$ be the solution obtained before the $k$th epoch, which is initialized to be $\mathbf{0}$ for the first epoch. Instead of searching for $\mathbf{w}_*$ at the $k$th epoch, our goal is to find $\mathbf{w}_* - \bar{\mathbf{w}}_k$, resulting in the following optimization problem for the $k$th epoch

$$\min_{\substack{\mathbf{w} + \mathbf{w}_k \in \mathcal{W} \\ \|\mathbf{w}\| \le \Delta_k}} \frac{\lambda_k}{2}\|\mathbf{w} + \bar{\mathbf{w}}_k\|^2 + \frac{1}{n}\sum_{i=1}^{n} g_i(\mathbf{w} + \bar{\mathbf{w}}_k), \tag{2}$$

where $\Delta_k$ specifies the domain size of $\mathbf{w}$ and $\lambda_k$ is the regularization parameter introduced at the $k$th epoch. By introducing the $\ell_2$ regularizer, the objective function in (2) becomes strongly convex, making it possible to exploit the technique for stochastic optimization of strongly convex function in order to improve the convergence rate. The domain size $\Delta_k$ and the regularization parameter $\lambda_k$ are initialized to be $\Delta_1 > 0$ and $\lambda_1 > 0$, respectively, and are reduced by a constant factor $\gamma > 1$ every epoch, i.e., $\Delta_k = \Delta_1/\gamma^{k-1}$ and $\lambda_k = \lambda_1/\gamma^{k-1}$. By removing the constant term $\lambda_k\|\bar{\mathbf{w}}_k\|^2/2$ from the objective function in (2), we obtain the following optimization problem for the $k$th epoch

$$\min_{\mathbf{w} \in \mathcal{W}_k} \left[ \mathcal{F}_k(\mathbf{w}) = \frac{\lambda_k}{2}\|\mathbf{w}\|^2 + \lambda_k \langle \mathbf{w}, \bar{\mathbf{w}}_k \rangle + \frac{1}{n}\sum_{i=1}^{n} g_i(\mathbf{w} + \bar{\mathbf{w}}_k) \right], \tag{3}$$

where $\mathcal{W}_k = \{\mathbf{w} : \mathbf{w} + \mathbf{w}_k \in \mathcal{W}, \ \|\mathbf{w}\| \le \Delta_k\}$. We rewrite the objective function $\mathcal{F}_k(\mathbf{w})$ as

$$
\begin{aligned}
\mathcal{F}_k(\mathbf{w}) &= \frac{\lambda_k}{2}\|\mathbf{w}\|^2 + \lambda_k\langle \mathbf{w}, \bar{\mathbf{w}}_k\rangle + \frac{1}{n}\sum_{i=1}^n g_i(\mathbf{w} + \bar{\mathbf{w}}_k) \\
&= \frac{\lambda_k}{2}\|\mathbf{w}\|^2 + \left\langle \mathbf{w}, \lambda_k\bar{\mathbf{w}}_k + \frac{1}{n}\sum_{i=1}^n \nabla g_i(\bar{\mathbf{w}}_k)\right\rangle + \frac{1}{n}\sum_{i=1}^n g_i(\mathbf{w} + \bar{\mathbf{w}}_k) - \langle \mathbf{w}, \nabla g_i(\bar{\mathbf{w}}_k)\rangle \\
&= \frac{\lambda_k}{2}\|\mathbf{w}\|^2 + \langle \mathbf{w}, \mathbf{g}_k\rangle + \frac{1}{n}\sum_{i=1}^n \widehat{g}_i^k(\mathbf{w}) \quad\quad\quad (4)
\end{aligned}
$$

where

$$
\mathbf{g}_k = \lambda_k\bar{\mathbf{w}}_k + \frac{1}{n}\sum_{i=1}^n \nabla g_i(\bar{\mathbf{w}}_k) \ \text{ and } \ \widehat{g}_i^k(\mathbf{w}) = g_i(\mathbf{w} + \bar{\mathbf{w}}_k) - \langle \mathbf{w}, \nabla g_i(\bar{\mathbf{w}}_k)\rangle.
$$

The main reason for using $\widehat{g}_i^k(\mathbf{w})$ instead of $g_i(\mathbf{w})$ is to tolerate the variance in the stochastic gradients. To see this, from the smoothness assumption of $g_i(\mathbf{w})$ we obtain the following inequality for the norm of $\widehat{g}_i^k(\mathbf{w})$ as:

$$
\left\|\nabla\widehat{g}_i^k(\mathbf{w})\right\| = \|\nabla g_i(\mathbf{w} + \bar{\mathbf{w}}_k) - \nabla g_i(\bar{\mathbf{w}}_k)\| \le \beta\|\mathbf{w}\|.
$$

As a result, since $\|\mathbf{w}\| \le \Delta_k$ and $\Delta_k$ shrinks over epochs, then $\|\mathbf{w}\|$ will approach to zero over epochs and consequentially $\|\nabla\widehat{g}_i^k(\mathbf{w})\|$ approaches to zero, which allows us to effectively control the variance in stochastic gradients, a key to improving the convergence of stochastic optimization for smooth functions to $O(1/T)$.

Using $\mathcal{F}_k(\mathbf{w})$ in (4), at the $t$th iteration of the $k$th epoch, we call the stochastic oracle $\mathcal{O}_s$ to randomly select a loss function $g_{i_t^k}(\mathbf{w})$ and update the solution by following the standard paradigm of SGD by

$$
\begin{aligned}
\mathbf{w}_k^{t+1} &= \Pi_{\mathbf{w}\in\mathcal{W}_k}\left(\mathbf{w}_k^t - \eta_k(\lambda_k\mathbf{w}_k^t + \mathbf{g}_k + \nabla\widehat{g}_{i_k^t}^k(\mathbf{w}_k^t))\right) \\
&= \Pi_{\mathbf{w}\in\mathcal{W}_k}\left(\mathbf{w}_k^t - \eta_k(\lambda_k\mathbf{w}_k^t + \mathbf{g}_k + \nabla g_{i_k^t}(\mathbf{w}_k^t + \bar{\mathbf{w}}_k) - \nabla g_{i_k^t}(\bar{\mathbf{w}}_k))\right), \quad\quad (5)
\end{aligned}
$$

where $\Pi_{\mathbf{w}\in\mathcal{W}_k}(\mathbf{w})$ projects the solution $\mathbf{w}$ into the domain $\mathcal{W}_k$ that shrinks over epochs.

At the end of each epoch, we compute the average solution $\widetilde{\mathbf{w}}_k$, and update the solution from $\bar{\mathbf{w}}_k$ to $\bar{\mathbf{w}}_{k+1} = \bar{\mathbf{w}}_k + \widetilde{\mathbf{w}}_k$. Similar to the epoch gradient descent algorithm [11], we increase the number of iterations by a constant $\gamma^2$ for every epoch, i.e. $T_k = T_1\gamma^{2(k-1)}$.

In order to perform stochastic gradient updating given in (5), we need to compute vector $\mathbf{g}_k$ at the beginning of the $k$th epoch, which requires an access to the full gradient oracle $\mathcal{O}_f$. It is easy to count that the number of accesses to the full gradient oracle $\mathcal{O}_f$ is $m$, and the number of accesses to the stochastic oracle $\mathcal{O}_s$ is

$$
T = T_1\sum_{i=1}^m \gamma^{2(i-1)} = \frac{\gamma^{2m} - 1}{\gamma^2 - 1}T_1.
$$

Thus, if the total number of accesses to the stochastic gradient oracle is $T$, the number of access to the full gradient oracle required by Algorithm 1 is $O(\ln T)$, consistent with our goal of making a small number of calls to the full gradient oracle.

The theorem below shows that for smooth objective functions, by having $O(\ln T)$ access to the full gradient oracle $\mathcal{O}_f$ and $O(T)$ access to the stochastic oracle $\mathcal{O}_s$, by running MIXEDGRAD algorithm, we achieve an optimization error of $O(1/T)$.

**Theorem 1.** *Let $\delta \le e^{-9/2}$ be the failure probability. Set $\gamma = 2$, $\lambda_1 = 16\beta$ and*

$$
T_1 = 300\ln\frac{m}{\delta}, \quad \eta_1 = \frac{1}{2\beta\sqrt{3T_1}}, \ \text{ and } \ \Delta_1 = R.
$$

*Define $T = T_1\left(2^{2m} - 1\right)/3$. Let $\bar{\mathbf{w}}_{m+1}$ be the solution returned by Algorithm 1 after $m$ epochs with $m = O(\ln T)$ calls to the full gradient oracle $\mathcal{O}_f$ and $T$ calls to the stochastic oracle $\mathcal{O}_s$. Then, with a probability $1 - 2\delta$, we have*

$$
\mathcal{G}(\bar{\mathbf{w}}_{m+1}) - \min_{\mathbf{w}\in\mathcal{W}}\mathcal{G}(\mathbf{w}) \le \frac{80\beta R^2}{2^{2m-2}} = O\left(\frac{\beta}{T}\right).
$$

# 5 Convergence Analysis

Now we turn to proving the main theorem. The proof will be given in a series of lemmas and theorems where the proof of few are given in the Appendix. The proof of main theorem is based on induction. To this end, let $\widehat{\mathbf{w}}_*^k$ be the optimal solution that minimizes $\mathcal{F}_k(\mathbf{w})$ defined in (3). The key to our analysis is show that when $\|\widehat{\mathbf{w}}_*^k\| \leq \Delta_k$, with a high probability, it holds that $\|\widehat{\mathbf{w}}_*^{k+1}\| \leq \Delta_k/\gamma$, where $\widehat{\mathbf{w}}_*^{k+1}$ is the optimal solution that minimizes $\mathcal{F}_{k+1}(\mathbf{w})$, as revealed by the following theorem.

**Theorem 2.** *Let $\widehat{\mathbf{w}}_*^k$ and $\widehat{\mathbf{w}}_*^{k+1}$ be the optimal solutions that minimize $\mathcal{F}_k(\mathbf{w})$ and $\mathcal{F}_{k+1}(\mathbf{w})$, respectively, and $\widetilde{\mathbf{w}}_{k+1}$ be the average solution obtained at the end of kth epoch of* MIXEDGRAD *algorithm. Suppose $\|\widehat{\mathbf{w}}_*^k\| \leq \Delta_k$. By setting the step size $\eta_k = 1/\left(2\beta\sqrt{3T_k}\right)$, we have, with a probability $1 - 2\delta$,*

$$\|\widehat{\mathbf{w}}_*^{k+1}\| \leq \frac{\Delta_k}{\gamma} \ \ and \ \ \mathcal{F}_k(\widetilde{\mathbf{w}}_{k+1}) - \min_{\mathbf{w}} \mathcal{F}_k(\mathbf{w}) \leq \frac{\lambda_k \Delta_k^2}{2\gamma^4}$$

*provided that $\delta \leq e^{-9/2}$ and*

$$T_k \geq \frac{300\gamma^8\beta^2}{\lambda_k^2} \ln\frac{1}{\delta}.$$

Taking this statement as given for the moment, we proceed with the proof of Theorem 1, returning later to establish the claim stated in Theorem 2.

*Proof of Theorem 1.* It is easy to check that for the first epoch, using the fact $\mathcal{W} \in \mathbb{B}_R$, we have

$$\|\mathbf{w}_*^1\| = \|\mathbf{w}_*\| \leq R := \Delta_1.$$

Let $\mathbf{w}_*^m$ be the optimal solution that minimizes $\mathcal{F}_m(\mathbf{w})$ and let $\widehat{\mathbf{w}}_*^{m+1}$ be the optimal solution obtained in the last epoch. Using Theorem 2, with a probability $1 - 2m\delta$, we have

$$\|\widehat{\mathbf{w}}_*^m\| \leq \frac{\Delta_1}{\gamma^{m-1}}, \quad \mathcal{F}_m(\widetilde{\mathbf{w}}_{m+1}) - \mathcal{F}_m(\widehat{\mathbf{w}}_*^m) \leq \frac{\lambda_m \Delta_m^2}{2\gamma^4} = \frac{\lambda_1 \Delta_1^2}{2\gamma^{3m+1}}$$

Hence by expanding the left hand side and utilizing the smoothness of individual loss functions we get

$$\begin{aligned} \frac{1}{n}\sum_{i=1}^n g_i(\bar{\mathbf{w}}_{m+1}) &\leq \mathcal{F}_m(\widehat{\mathbf{w}}_*^m) + \frac{\lambda_1 \Delta_1^2}{2\gamma^{3m+1}} - \frac{\lambda_1}{\gamma^{m-1}}\langle \widetilde{\mathbf{w}}_{m+1}, \bar{\mathbf{w}}_m \rangle \\ &\leq \mathcal{F}_m(\widehat{\mathbf{w}}_*^m) + \frac{\lambda_1 \Delta_1^2}{2\gamma^{3m+1}} + \frac{\lambda_1 \|\bar{\mathbf{w}}_m\|\Delta_1}{\gamma^{2m-2}} \end{aligned}$$

where the last step uses the fact $\|\widehat{\mathbf{w}}_*^{m+1}\| \leq \Delta_m = \Delta_1\gamma^{1-m}$. Since

$$\|\bar{\mathbf{w}}_m\| \leq \sum_{i=1}^m |\widetilde{\mathbf{w}}_i| \leq \sum_{i=1}^m \Delta_i \leq \frac{\gamma\Delta_1}{\gamma - 1} \leq 2\Delta_1$$

where in the last step holds under the condition $\gamma \geq 2$. By combining above inequalities, we obtain

$$\frac{1}{n}\sum_{i=1}^n g_i(\bar{\mathbf{w}}_{m+1}) \leq \mathcal{F}_m(\widehat{\mathbf{w}}_*^m) + \frac{\lambda_1 \Delta_1^2}{2\gamma^{3m+1}} + \frac{2\lambda_1 \Delta_1^2}{\gamma^{2m-2}}.$$

Our final goal is to relate $\mathcal{F}_m(\mathbf{w})$ to $\min_{\mathbf{w}} \mathcal{G}(\mathbf{w})$. Since $\widehat{\mathbf{w}}_*^m$ minimizes $\mathcal{F}_m(\mathbf{w})$, for any $\mathbf{w}_* \in \arg\min \mathcal{G}(\mathbf{w})$, we have

$$\mathcal{F}_m(\mathbf{w}_*^m) \leq \mathcal{F}_m(\mathbf{w}_*) = \frac{1}{n}\sum_{i=1}^n g_i(\mathbf{w}_*) + \frac{\lambda_1}{2\gamma^{m-1}}\left(\|\mathbf{w}_* - \bar{\mathbf{w}}_m\|^2 + 2\langle \mathbf{w}_* - \bar{\mathbf{w}}_m, \bar{\mathbf{w}}_m \rangle\right). \quad (6)$$

Thus, the key to bound $|\mathcal{F}(\mathbf{w}_*^m) - \mathcal{G}(\mathbf{w}_*)|$ is to bound $\|\mathbf{w}_* - \bar{\mathbf{w}}_m\|$. To this end, after the first $m$ epoches, we run Algorithm 1 with *full gradients*. Let $\bar{\mathbf{w}}_{m+1}, \bar{\mathbf{w}}_{m+2}, \ldots$ be the sequence of solutions generated by Algorithm 1 after the first $m$ epoches. For this sequence of solutions, Theorem 2 will hold deterministically as we deploy the full gradient for updating, i.e., $\|\widetilde{\mathbf{w}}_k\| \leq \Delta_k$ for any $k \geq m + 1$. Since we reduce $\lambda_k$ exponentially, $\lambda_k$ will approach to zero and therefore the sequence $\{\bar{\mathbf{w}}_k\}_{k=m+1}^\infty$ will converge to $\mathbf{w}_*$, one of the optimal solutions that minimize $\mathcal{G}(\mathbf{w})$. Since $\mathbf{w}_*$ is the limit of sequence $\{\bar{\mathbf{w}}_k\}_{k=m+1}^\infty$ and $\|\bar{\mathbf{w}}_k\| \leq \Delta_k$ for any $k \geq m + 1$, we have

$$\|\mathbf{w}_* - \bar{\mathbf{w}}_m\| \leq \sum_{i=m+1}^\infty |\widetilde{\mathbf{w}}_i| \leq \sum_{k=m+1}^\infty \Delta_k \leq \frac{\Delta_1}{\gamma^m(1 - \gamma^{-1})} \leq \frac{2\Delta_1}{\gamma^m}$$

where the last step follows from the condition $\gamma \geq 2$. Thus,

$$
\begin{aligned}
\mathcal{F}_m(\mathbf{w}_*^m) &\leq \frac{1}{n}\sum_{i=1}^n g_i(\mathbf{w}_*) + \frac{\lambda_1}{2\gamma^{m-1}}\left(\frac{4\Delta_1^2}{\gamma^{2m}} + \frac{8\Delta_1^2}{\gamma^m}\right) \\
&= \frac{1}{n}\sum_{i=1}^n g_i(\mathbf{w}_*) + \frac{2\lambda_1\Delta_1^2}{\gamma^{2m-1}}\left(2+\gamma^{-m}\right) \leq \frac{1}{n}\sum_{i=1}^n g_i(\mathbf{w}_*) + \frac{5\lambda_1\Delta_1^2}{\gamma^{2m-1}} \quad (7)
\end{aligned}
$$

By combining the bounds in (6) and (7), we have, with a probability $1 - 2m\delta$,

$$
\frac{1}{n}\sum_{i=1}^n g_i(\bar{\mathbf{w}}_{m+1}) - \frac{1}{n}\sum_{i=1}^n g_i(\mathbf{w}_*) \leq \frac{5\lambda_1\Delta_1^2}{\gamma^{2m-2}} = O(1/T)
$$

where

$$
T = T_1\sum_{k=0}^{m-1}\gamma^{2k} = \frac{T_1\left(\gamma^{2m}-1\right)}{\gamma^2-1} \leq \frac{T_1}{3}\gamma^{2m}.
$$

We complete the proof by plugging in the stated values for $\gamma$, $\lambda_1$ and $\Delta_1$. $\qquad\square$

## 5.1 Proof of Theorem 2

For the convenience of discussion, we drop the subscript $k$ for epoch just to simplify our notation. Let $\lambda = \lambda_k$, $T = T_k$, $\Delta = \Delta_k$, $\mathbf{g} = \mathbf{g}_k$. Let $\bar{\mathbf{w}} = \bar{\mathbf{w}}_k$ be the solution obtained before the start of the epoch $k$, and let $\bar{\mathbf{w}}' = \bar{\mathbf{w}}_{k+1}$ be the solution obtained after running through the $k$th epoch. We denote by $\mathcal{F}(\mathbf{w})$ and $\mathcal{F}'(\mathbf{w})$ the objective functions $\mathcal{F}_k(\mathbf{w})$ and $\mathcal{F}_{k+1}(\mathbf{w})$. They are given by

$$
\mathcal{F}(\mathbf{w}) = \frac{\lambda}{2}\|\mathbf{w}\|^2 + \lambda\langle\mathbf{w}, \bar{\mathbf{w}}\rangle + \frac{1}{n}\sum_{i=1}^n g_i(\mathbf{w}+\bar{\mathbf{w}}) \quad (8)
$$

$$
\mathcal{F}'(\mathbf{w}) = \frac{\lambda}{2\gamma}\|\mathbf{w}\|^2 + \frac{\lambda}{\gamma}\langle\mathbf{w}, \bar{\mathbf{w}}'\rangle + \frac{1}{n}\sum_{i=1}^n g_i(\mathbf{w}+\bar{\mathbf{w}}') \quad (9)
$$

Let $\widehat{\mathbf{w}}_* = \widehat{\mathbf{w}}_*^k$ and $\widehat{\mathbf{w}}_*' = \widehat{\mathbf{w}}_*^{k+1}$ be the optimal solutions that minimize $\mathcal{F}(\mathbf{w})$ and $\mathcal{F}'(\mathbf{w})$ over the domain $\mathcal{W}_k$ and $\mathcal{W}_{k+1}$, respectively. Under the assumption that $\|\widehat{\mathbf{w}}_*\| \leq \Delta$, our goal is to show

$$
\|\widehat{\mathbf{w}}_*'\| \leq \frac{\Delta}{\gamma}, \quad \mathcal{F}(\bar{\mathbf{w}}') - \mathcal{F}(\widehat{\mathbf{w}}_*) \leq \frac{\lambda\Delta^2}{2\gamma^4}
$$

The following lemma bounds $\mathcal{F}(\mathbf{w}_t) - \mathcal{F}(\widehat{\mathbf{w}}_*)$ where the proof is deferred to Appendix.

**Lemma 1.**

$$
\begin{aligned}
\mathcal{F}(\mathbf{w}_t) - \mathcal{F}(\widehat{\mathbf{w}}_*) &\leq \frac{\|\mathbf{w}_t - \widehat{\mathbf{w}}_*\|^2}{2\eta} - \frac{\|\mathbf{w}_{t+1} - \widehat{\mathbf{w}}_*\|^2}{2\eta} + \frac{\eta}{2}\|\nabla\widehat{g}_{i_t}(\mathbf{w}_t) + \lambda\mathbf{w}_t\|^2 + \langle\mathbf{g}, \mathbf{w}_t - \mathbf{w}_{t+1}\rangle \\
&\quad + \left\langle\nabla\widehat{\mathcal{F}}(\widehat{\mathbf{w}}_*) - \nabla\widehat{g}_{i_t}(\widehat{\mathbf{w}}_*), \mathbf{w}_t - \widehat{\mathbf{w}}_*\right\rangle + \left\langle-\nabla\widehat{g}_{i_t}(\mathbf{w}_t) + \nabla\widehat{g}_{i_t}(\widehat{\mathbf{w}}_*) - \nabla\widehat{\mathcal{F}}(\widehat{\mathbf{w}}_*) + \nabla\widehat{\mathcal{F}}(\mathbf{w}_t), \mathbf{w}_t - \widehat{\mathbf{w}}_*\right\rangle
\end{aligned}
$$

By adding the inequality in Lemma 1 over all iterations, using the fact $\bar{\mathbf{w}}_1 = \mathbf{0}$, we have

$$
\begin{aligned}
\sum_{t=1}^T \mathcal{F}(\mathbf{w}_t) - \mathcal{F}(\widehat{\mathbf{w}}_*) &\leq \frac{\|\widehat{\mathbf{w}}_*\|^2}{2\eta} - \frac{\|\mathbf{w}_{T+1} - \widehat{\mathbf{w}}_*\|^2}{2\eta} - \langle\mathbf{g}, \mathbf{w}_{T+1}\rangle \\
&\quad + \underbrace{\frac{\eta}{2}\sum_{t=1}^T\|\nabla\widehat{g}_{i_t}(\mathbf{w}_t) + \lambda\mathbf{w}_t\|^2}_{:=A_T} + \underbrace{\sum_{t=1}^T\langle\nabla\widehat{\mathcal{F}}(\widehat{\mathbf{w}}_*) - \nabla\widehat{g}_{i_t}(\widehat{\mathbf{w}}_*), \mathbf{w}_t - \widehat{\mathbf{w}}_*\rangle}_{:=B_T} \\
&\quad + \underbrace{\sum_{t=1}^T\left\langle-\nabla\widehat{g}_{i_t}(\mathbf{w}_t) + \nabla\widehat{g}_{i_t}(\widehat{\mathbf{w}}_*) - \nabla\widehat{\mathcal{F}}(\widehat{\mathbf{w}}_*) + \nabla\widehat{\mathcal{F}}(\mathbf{w}_t), \mathbf{w}_t - \widehat{\mathbf{w}}_*\right\rangle}_{:=C_T}.
\end{aligned}
$$

Since $\mathbf{g} = \nabla\mathcal{F}(\mathbf{0})$ and

$$
\mathcal{F}(\mathbf{w}_{T+1}) - \mathcal{F}(\mathbf{0}) \leq \langle\nabla\mathcal{F}(\mathbf{0}), \mathbf{w}_{T+1}\rangle + \frac{\beta}{2}\|\mathbf{w}_{T+1}\|^2 = \langle\mathbf{g}, \mathbf{w}_{T+1}\rangle + \frac{\beta}{2}\|\mathbf{w}_{T+1}\|^2
$$

using the fact $\mathcal{F}(\mathbf{0}) \leq \mathcal{F}(\mathbf{w}_*) + \frac{\beta}{2}\|\mathbf{w}_*\|^2$ and $\max(\|\mathbf{w}_*\|, \|\mathbf{w}_{T+1}\|) \leq \Delta$, we have

$$-\langle \mathbf{g}, \mathbf{w}_{T+1} \rangle \leq \mathcal{F}(\mathbf{0}) - \mathcal{F}(\mathbf{w}_{T+1}) + \frac{\beta}{2}\Delta^2 \leq \beta\Delta^2 - (\mathcal{F}(\mathbf{w}_{T+1}) - \mathcal{F}(\widehat{\mathbf{w}}_*))$$

and therefore

$$\sum_{t=1}^{T+1} \mathcal{F}(\mathbf{w}_t) - \mathcal{F}(\widehat{\mathbf{w}}_*) \leq \Delta^2 \left( \frac{1}{2\eta} + \beta \right) + \frac{\eta}{2}A_T + B_T + C_T. \tag{10}$$

The following lemmas bound $A_T$, $B_T$ and $C_T$.

**Lemma 2.** *For $A_T$ defined above we have $A_T \leq 6\beta^2\Delta^2 T$.*

The following lemma upper bounds $B_T$ and $C_T$. The proof is based on the Bernstein's inequality for Martingales [4] and is given in the Appendix.

**Lemma 3.** *With a probability $1 - 2\delta$, we have*

$$B_T \leq \beta\Delta^2 \left( \ln\frac{1}{\delta} + \sqrt{2T\ln\frac{1}{\delta}} \right) \quad and \quad C_T \leq 2\beta\Delta^2 \left( \ln\frac{1}{\delta} + \sqrt{2T\ln\frac{1}{\delta}} \right).$$

Using Lemmas 2 and 3, by substituting the uppers bounds for $A_T$, $B_T$, and $C_T$ in (10), with a probability $1 - 2\delta$, we obtain

$$\sum_{t=1}^{T+1} \mathcal{F}(\mathbf{w}_t) - \mathcal{F}(\widehat{\mathbf{w}}_*) \leq \Delta^2 \left( \frac{1}{2\eta} + \beta + 6\beta^2\eta T + 3\beta\ln\frac{1}{\delta} + 3\beta\sqrt{2T\ln\frac{1}{\delta}} \right)$$

By choosing $\eta = 1/[2\beta\sqrt{3T}]$, we have

$$\sum_{t=1}^{T+1} \mathcal{F}(\mathbf{w}_t) - \mathcal{F}(\widehat{\mathbf{w}}_*) \leq \Delta^2 \left( 2\beta\sqrt{3T} + \beta + 3\beta\ln\frac{1}{\delta} + 3\beta\sqrt{2T\ln\frac{1}{\delta}} \right)$$

and using the fact $\widetilde{\mathbf{w}} = \sum_{i=1}^{T+1} \mathbf{w}_t/(T+1)$, we have

$$\mathcal{F}(\widetilde{\mathbf{w}}) - \mathcal{F}(\widehat{\mathbf{w}}_*) \leq \Delta^2 \frac{5\beta\sqrt{3\ln[1/\delta]}}{\sqrt{T+1}}, \quad \text{and} \quad \widehat{\Delta}^2 = \|\widetilde{\mathbf{w}} - \widehat{\mathbf{w}}_*\|^2 \leq \Delta^2 \frac{5\beta\sqrt{3\ln[1/\delta]}}{\lambda\sqrt{T+1}}.$$

Thus, when $T \geq [300\gamma^8\beta^2 \ln\frac{1}{\delta}]/\lambda^2$, we have, with a probability $1 - 2\delta$,

$$\widehat{\Delta}^2 \leq \frac{\Delta^2}{\gamma^4}, \quad \text{and} \quad |\mathcal{F}(\widetilde{\mathbf{w}}) - \mathcal{F}(\widehat{\mathbf{w}}_*)| \leq \frac{\lambda}{2\gamma^4}\Delta^2. \tag{11}$$

The next lemma relates $\|\widehat{\mathbf{w}}_*'\|$ to $\|\widetilde{\mathbf{w}} - \widehat{\mathbf{w}}_*\|$.

**Lemma 4.** *We have $\|\widehat{\mathbf{w}}_*'\| \leq \gamma\|\widetilde{\mathbf{w}} - \widehat{\mathbf{w}}_*\|$.*

Combining the bound in (11) with Lemma 4, we have $\|\widehat{\mathbf{w}}_*'\| \leq \Delta/\gamma$.

## 6 Conclusions and Open Questions

We presented a new paradigm of optimization, termed as mixed optimization, that aims to improve the convergence rate of stochastic optimization by making a small number of calls to the full gradient oracle. We proposed the MIXEDGRAD algorithm and showed that it is able to achieve an $O(1/T)$ convergence rate by accessing stochastic and full gradient oracles for $\mathcal{O}(T)$ and $\mathcal{O}(\log T)$ times, respectively. We showed that the MIXEDGRAD algorithm is able to exploit the *smoothness* of the function, which is believed to be not very useful in stochastic optimization.

In the future, we would like to examine the optimality of our algorithm, namely if it is possible to achieve a better convergence rate for stochastic optimization of smooth functions using $O(\ln T)$ accesses to the full gradient oracle. Furthermore, to alleviate the computational cost caused by $O(\log T)$ accesses to the full gradient oracle, it would be interesting to empirically evaluate the proposed algorithm in a distributed framework by distributing the individual functions among processors to parallelize the full gradient computation at the beginning of each epoch which requires $O(\log T)$ communications between the processors in total. Lastly, it is very interesting to check whether an $O(1/T^2)$ rate could be achieved by an accelerated method in the mixed optimization scenario, and whether linear convergence rates could be achieved in the strongly-convex case.

**Acknowledgments.** The authors would like to thank the anonymous reviewers for their helpful and insightful comments. This work was supported in part by ONR Award N000141210431 and NSF (IIS-1251031).

## Footnotes

[1]The convergence rate can be improved to $O(1/T)$ when the structure of the objective function is provided.

[2]We note that the stochastic oracle assumed in our study is slightly stronger than the stochastic gradient oracle as it returns the sampled function instead of the stochastic gradient.

[3]The audience may feel that this relaxation of stochastic oracle could provide significantly more information, and second order methods such as Online Newton [10] may be applied to achieve $O(1/T)$ convergence. We note (i) the proposed algorithm is a first order method, and (ii) although the Online Newton method yields a regret bound of $O(1/T)$, its convergence rate for optimization can be as low as $O(1/\sqrt{T})$ due to the concentration bound for Martingales. In addition, the Online Newton method is only applicable to exponential concave function, not any smooth loss function.

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
