[Supplementary Material]

# Supplementary Material for
# Mixed Optimization for Smooth Functions

**Mehrdad Mahdavi**    **Lijun Zhang**    **Rong Jin**

Department of Computer Science and Engineering, Michigan State University, MI, USA

{mahdavim,zhanglij,rongjin}@msu.edu

Before proving the lemmas we recall the definition of $\mathcal{F}(\mathbf{w})$, $\mathcal{F}'(\mathbf{w})$, $\mathbf{g}$, and $\widehat{g}_i(\mathbf{w})$ as:

$$\mathcal{F}(\mathbf{w}) = \frac{\lambda}{2}\|\mathbf{w}\|^2 + \lambda\langle \mathbf{w}, \bar{\mathbf{w}} \rangle + \frac{1}{n}\sum_{i=1}^{n} g_i(\mathbf{w} + \bar{\mathbf{w}}),$$

$$\mathcal{F}'(\mathbf{w}) = \frac{\lambda}{2\gamma}\|\mathbf{w}\|^2 + \frac{\lambda}{\gamma}\langle \mathbf{w}, \bar{\mathbf{w}}' \rangle + \frac{1}{n}\sum_{i=1}^{n} g_i(\mathbf{w} + \bar{\mathbf{w}}'),$$

$$\mathbf{g} = \lambda\bar{\mathbf{w}} + \frac{1}{n}\sum_{i=1}^{n} \nabla g_i(\bar{\mathbf{w}}),$$

$$\widehat{g}_i(\mathbf{w}) = g_i(\mathbf{w} + \bar{\mathbf{w}}) - \langle \mathbf{w}, \nabla g_i(\bar{\mathbf{w}}) \rangle.$$

We also recall that $\widehat{\mathbf{w}}_*$ and $\widehat{\mathbf{w}}'_*$ are the optimal solutions that minimize $\mathcal{F}(\mathbf{w})$ and $\mathcal{F}'(\mathbf{w})$ over the domain $\mathcal{W}_k$ and $\mathcal{W}_{k+1}$, respectively.

**Lemma 1.**

$$\mathcal{F}(\mathbf{w}_t) - \mathcal{F}(\widehat{\mathbf{w}}_*) \leq \frac{\|\mathbf{w}_t - \widehat{\mathbf{w}}_*\|^2}{2\eta} - \frac{\|\mathbf{w}_{t+1} - \widehat{\mathbf{w}}_*\|^2}{2\eta} + \frac{\eta}{2}\left\|\nabla \widehat{g}_{i_t}(\mathbf{w}_t) + \lambda\mathbf{w}_t\right\|^2 + \langle \mathbf{g}, \mathbf{w}_t - \mathbf{w}_{t+1} \rangle$$

$$+ \left\langle \nabla \widehat{g}_{i_t}(\widehat{\mathbf{w}}_*) - \nabla \widehat{\mathcal{F}}(\widehat{\mathbf{w}}_*), \mathbf{w}_t - \widehat{\mathbf{w}}_* \right\rangle + \left\langle -\nabla \widehat{g}_{i_t}(\mathbf{w}_t) + \nabla \widehat{g}_{i_t}(\widehat{\mathbf{w}}_*) - \nabla \widehat{\mathcal{F}}(\widehat{\mathbf{w}}_*) + \nabla \widehat{\mathcal{F}}(\mathbf{w}_t), \mathbf{w}_t - \widehat{\mathbf{w}}_* \right\rangle$$

*Proof.* For each iteration $t$ in the $k$th epoch, from the strong convexity of $\mathcal{F}(\mathbf{w})$ we have

$$\mathcal{F}(\mathbf{w}_t) - \mathcal{F}(\widehat{\mathbf{w}}_*) \leq \langle \nabla \mathcal{F}(\mathbf{w}_t), \mathbf{w}_t - \widehat{\mathbf{w}}_* \rangle - \frac{\lambda}{2}\|\mathbf{w}_t - \widehat{\mathbf{w}}_*\|^2$$

$$= \langle \mathbf{g} + \nabla \widehat{g}_{i_t}(\mathbf{w}_t) + \lambda\mathbf{w}_t, \mathbf{w}_t - \widehat{\mathbf{w}}_* \rangle + \left\langle -\nabla \widehat{g}_{i_t}(\mathbf{w}_t) + \nabla \widehat{\mathcal{F}}(\mathbf{w}_t), \mathbf{w}_t - \widehat{\mathbf{w}}_* \right\rangle - \frac{\lambda}{2}\|\mathbf{w}_t - \widehat{\mathbf{w}}_*\|^2,$$

where $\widehat{\mathcal{F}}(\mathbf{w}) = \frac{1}{n}\sum_{i=1}^{n} \widehat{g}_i(\mathbf{w})$. We now try to upper bound the first term in the right hand side. Since

$$\langle \mathbf{g} + \nabla \widehat{g}_{i_t}(\mathbf{w}_t) + \lambda\mathbf{w}_t, \mathbf{w}_t - \widehat{\mathbf{w}}_* \rangle$$

$$= \langle \mathbf{g} + \nabla \widehat{g}_{i_t}(\mathbf{w}_t) + \lambda\mathbf{w}_t, \mathbf{w}_t - \widehat{\mathbf{w}}_* \rangle - \frac{\|\mathbf{w}_t - \widehat{\mathbf{w}}_*\|^2}{2\eta} + \frac{\|\mathbf{w}_t - \widehat{\mathbf{w}}_*\|^2}{2\eta}$$

$$\leq \langle \mathbf{g} + \nabla \widehat{g}_{i_t}(\mathbf{w}_t) + \lambda\mathbf{w}_t, \mathbf{w}_t - \mathbf{w}_{t+1} \rangle - \frac{\|\mathbf{w}_t - \mathbf{w}_{t+1}\|^2}{2\eta} - \frac{\|\mathbf{w}_{t+1} - \widehat{\mathbf{w}}_*\|^2}{2\eta} + \frac{\|\mathbf{w}_t - \widehat{\mathbf{w}}_*\|^2}{2\eta}$$

$$\leq \langle \mathbf{g}, \mathbf{w}_t - \mathbf{w}_{t+1} \rangle - \frac{\|\mathbf{w}_{t+1} - \widehat{\mathbf{w}}_*\|^2}{2\eta} + \frac{\|\mathbf{w}_t - \widehat{\mathbf{w}}_*\|^2}{2\eta} + \max_{\mathbf{w}}\left[\langle \nabla \widehat{g}_{i_t}(\mathbf{w}_t) + \lambda\mathbf{w}_t, \mathbf{w}_t - \mathbf{w} \rangle - \frac{\|\mathbf{w}_t - \mathbf{w}\|^2}{2\eta}\right]$$

$$= \langle \mathbf{g}, \mathbf{w}_t - \mathbf{w}_{t+1} \rangle - \frac{\|\mathbf{w}_{t+1} - \widehat{\mathbf{w}}_*\|^2}{2\eta} + \frac{\|\mathbf{w}_t - \widehat{\mathbf{w}}_*\|^2}{2\eta} + \frac{\eta}{2}\|\nabla \widehat{g}_{i_t}(\mathbf{w}_t) + \lambda\mathbf{w}_t\|^2$$

where the first inequality follows from the fact that $\mathbf{w}_{t+1}$ in the minimizer of the following optimization problem:

$$\mathbf{w}_{t+1} = \operatorname*{arg\,min}_{\mathbf{w}\in\mathcal{W}\wedge\|\mathbf{w}-\bar{\mathbf{w}}\|\leq\Delta} \langle \mathbf{g}+\nabla\widehat{g}_{i_t}(\mathbf{w}_t)+\lambda\mathbf{w}_t, \mathbf{w}-\mathbf{w}_t\rangle + \frac{\|\mathbf{w}-\mathbf{w}_t\|^2}{2\eta}.$$

Therefore, we obtain

$$
\begin{aligned}
\mathcal{F}(\mathbf{w}_t) &- \mathcal{F}(\widehat{\mathbf{w}}_*)\\
&\leq \frac{\|\mathbf{w}_t-\widehat{\mathbf{w}}_*\|^2}{2\eta} - \frac{\|\mathbf{w}_{t+1}-\widehat{\mathbf{w}}_*\|^2}{2\eta} - \frac{\lambda}{2}\|\mathbf{w}_t-\widehat{\mathbf{w}}_*\|^2\\
&\quad +\langle \mathbf{g}, \mathbf{w}_t-\mathbf{w}_{t+1}\rangle + \frac{\eta}{2}\|\nabla\widehat{g}_{i_t}(\mathbf{w}_t)+\lambda\mathbf{w}_t\|^2 + \left\langle \nabla\widehat{\mathcal{F}}(\widehat{\mathbf{w}}_*)-\nabla\widehat{g}_{i_t}(\widehat{\mathbf{w}}_*), \mathbf{w}_t-\widehat{\mathbf{w}}_*\right\rangle\\
&\quad + \left\langle -\nabla\widehat{g}_{i_t}(\mathbf{w}_t)+\nabla\widehat{g}_{i_t}(\widehat{\mathbf{w}}_*)-\nabla\widehat{\mathcal{F}}(\widehat{\mathbf{w}}_*)+\nabla\widehat{\mathcal{F}}(\mathbf{w}_t), \mathbf{w}_t-\widehat{\mathbf{w}}_*\right\rangle,
\end{aligned}
$$

as desired. $\qquad\square$

We now turn to prove the upper bound on $A_T$.

**Lemma 2.**
$$A_T \leq 6\beta^2\Delta^2 T$$

*Proof.* We bound $A_T$ as

$$
\begin{aligned}
A_T &= \sum_{t=1}^{T}\|\nabla\widehat{g}_{i_t}(\mathbf{w}_t)+\lambda\mathbf{w}_t\|^2\\
&\leq \sum_{t=1}^{T}2\|\nabla\widehat{g}_{i_t}(\mathbf{w}_t)\|^2 + 2\lambda^2\|\mathbf{w}_t\|^2\\
&\leq \sum_{t=1}^{T}2\lambda^2\Delta^2 + 2\|\nabla\widehat{g}_{i_t}(\mathbf{w}_t)-\nabla\widehat{g}_{i_t}(\widehat{\mathbf{w}}_*)+\nabla\widehat{g}_{i_t}(\widehat{\mathbf{w}}_*)\|^2 \leq 6\beta^2\Delta^2 T
\end{aligned}
$$

where the second inequality follows $(a+b)^2 \leq 2(a^2+b^2)$ and the last inequality follows from the smoothness assumption. $\qquad\square$

**Lemma 3.** *With a probability $1-2\delta$, we have*

$$B_T \leq \beta\Delta^2\left(\ln\frac{1}{\delta}+\sqrt{2T\ln\frac{1}{\delta}}\right) \quad\text{and}\quad C_T \leq 2\beta\Delta^2\left(\ln\frac{1}{\delta}+\sqrt{2T\ln\frac{1}{\delta}}\right)$$

The proof is based on the Berstein inequality for Martingales [1] which is restated here for completeness.

**Theorem 1.** *(Bernstein's inequality for martingales). Let $X_1,\dots,X_n$ be a bounded martingale difference sequence with respect to the filtration $\mathcal{F}=(\mathcal{F}_i)_{1\leq i\leq n}$ and with $\|X_i\|\leq K$. Let*

$$S_i = \sum_{j=1}^{i}X_j$$

*be the associated martingale. Denote the sum of the conditional variances by*

$$\Sigma_n^2 = \sum_{t=1}^{n}\mathbb{E}\left[X_t^2|\mathcal{F}_{t-1}\right],$$

*Then for all constants $t, \nu > 0$,*

$$\Pr\left[\max_{i=1,\dots,n}S_i > t \text{ and } \Sigma_n^2 \leq \nu\right] \leq \exp\left(-\frac{t^2}{2(\nu+Kt/3)}\right),$$

*and therefore,*

$$\Pr\left[\max_{i=1,\dots,n}S_i > \sqrt{2\nu t}+\frac{\sqrt{2}}{3}Kt \text{ and } \Sigma_n^2 \leq \nu\right] \leq e^{-t}.$$

Equipped with this theorem, we are now in a position to upper bound $B_T$ and $C_T$ as follows.

*Proof.* (of Lemma 3) Denote $X_t = \langle \nabla \widehat{g}_{i_t}(\widehat{\mathbf{w}}_*) - \nabla \widehat{\mathcal{F}}(\widehat{\mathbf{w}}_*), \mathbf{w}_t - \widehat{\mathbf{w}}_* \rangle$. We have that the conditional expectation of $X_t$, given randomness in previous rounds, is $\mathbb{E}_{t-1}[X_t] = 0$. We now apply Theorem 1 to the sum of martingale differences. In particular, we have, with a probability $1 - e^{-t}$,

$$B_T \leq \frac{\sqrt{2}}{3} K t + \sqrt{2\Sigma t}$$

where

$$K = \max_{1 \leq t \leq T} \langle \nabla \widehat{g}_{i_t}(\widehat{\mathbf{w}}_*) - \nabla \widehat{\mathcal{F}}(\widehat{\mathbf{w}}_*), \mathbf{w}_t - \widehat{\mathbf{w}}_* \rangle \leq 2\beta\Delta^2$$

$$\Sigma = \sum_{t=1}^{T} \mathbb{E}_t \left[ |\langle \nabla \widehat{g}_{i_t}(\widehat{\mathbf{w}}_*) - \nabla \widehat{\mathcal{F}}(\widehat{\mathbf{w}}_*), \mathbf{w}_t - \widehat{\mathbf{w}}_* \rangle|^2 \right] \leq \beta^2 \Delta^4 T$$

Hence, with a probability $1 - \delta$, we have

$$B_T \leq \beta\Delta^2 \left( \ln\frac{1}{\delta} + \sqrt{2T \ln\frac{1}{\delta}} \right)$$

Similar, for $C_T$, we have, with a probability $1 - \delta$,

$$C_T \leq 2\beta\Delta^2 \left( \ln\frac{1}{\delta} + \sqrt{2T \ln\frac{1}{\delta}} \right)$$

$\square$

**Lemma 4.** $\|\widehat{\mathbf{w}}'_*\| \leq \gamma \|\widetilde{\mathbf{w}} - \widehat{\mathbf{w}}_*\|$.

*Proof.* We rewrite $\mathcal{F}(\mathbf{w})$ as

$$\mathcal{F}(\mathbf{w}) = \frac{\lambda}{2}\|\mathbf{w}\|^2 + \lambda\langle \mathbf{w}, \bar{\mathbf{w}} \rangle + \frac{1}{n}\sum_{i=1}^{n} g_i(\mathbf{w} + \bar{\mathbf{w}})$$

$$= \frac{\lambda}{2}\|\mathbf{w} - \widetilde{\mathbf{w}} + \widetilde{\mathbf{w}}\|^2 + \lambda\langle \mathbf{w} - \widetilde{\mathbf{w}} + \widetilde{\mathbf{w}}, \bar{\mathbf{w}} \rangle + \frac{1}{n}\sum_{i=1}^{n} g_i(\mathbf{w} - \widetilde{\mathbf{w}} + \bar{\mathbf{w}}')$$

Define $\mathbf{z} = \mathbf{w} - \widetilde{\mathbf{w}}$. We have

$$\mathcal{F}(\mathbf{w}) = \frac{\lambda}{2}\|\mathbf{z} + \widetilde{\mathbf{w}}\|^2 + \lambda\langle \mathbf{z}, \bar{\mathbf{w}} \rangle + \lambda\langle \widetilde{\mathbf{w}}, \bar{\mathbf{w}} \rangle + \frac{1}{n}\sum_{i=1}^{n} g_i(\mathbf{z} + \bar{\mathbf{w}}')$$

$$= \frac{\lambda}{2}\|\mathbf{z}\|^2 + \lambda\langle \mathbf{z}, \bar{\mathbf{w}}' \rangle + \frac{1}{n}\sum_{i=1}^{n} g_i(\mathbf{z} + \bar{\mathbf{w}}') + \frac{\lambda}{2}\|\widetilde{\mathbf{w}}\|^2 + \lambda\langle \widetilde{\mathbf{w}}, \bar{\mathbf{w}} \rangle$$

$$= \widetilde{\mathcal{F}}(\mathbf{z}) + \frac{\lambda}{2}\|\widetilde{\mathbf{w}}\|^2 + \lambda\langle \widetilde{\mathbf{w}}, \bar{\mathbf{w}} \rangle$$

where

$$\widetilde{\mathcal{F}}(\mathbf{z}) = \frac{\lambda}{2}\|\mathbf{z}\|^2 + \lambda\langle \mathbf{z}, \bar{\mathbf{w}}' \rangle + \frac{1}{n}\sum_{i=1}^{n} g_i(\mathbf{z} + \bar{\mathbf{w}}')$$

Define $\widetilde{\mathbf{w}}_* = \widehat{\mathbf{w}}_* - \widetilde{\mathbf{w}}$. Evidently, $\widetilde{\mathbf{w}}_*$ minimizes $\widetilde{\mathcal{F}}(\mathbf{w})$. The only difference between $\widetilde{\mathcal{F}}(\mathbf{w})$ and $F'(\mathbf{w})$ is that they use different modulus of strong convexity $\lambda$. Thus, following [2], we have

$$\|\widetilde{\mathbf{w}}_* - \widehat{\mathbf{w}}'_*\| \leq \frac{1 - \gamma^{-1}}{\gamma^{-1}}\|\widetilde{\mathbf{w}}_*\| \leq (\gamma - 1)\|\widetilde{\mathbf{w}}_*\|$$

Hence,

$$\|\widehat{\mathbf{w}}'_*\| \leq \gamma\|\widetilde{\mathbf{w}}_*\| = \gamma\|\widehat{\mathbf{w}}_* - \widetilde{\mathbf{w}}\|$$

which completes the proofs. $\square$