[Reviews · NeurIPS 2013]

Submitted by Assigned_Reviewer_1

The paper considers optimization with a "mixed" oracle, which provides the algorithm access to the standard stochastic oracle as well as a small number of accesses to an exact oracle. In this setting, the authors give an algorithm that achieves a convergence rate of O(1/T) after O(T) calls to the stochastic oracle and O(log T) class to the exact oracle, improving on the known rates of O(1/sqrt(T)) after O(T) calls to the stochastic oracle and O(1/T) after O(T) calls to the exact oracle.

Comments:

The paper asks an interesting question, and provides an interesting answer to it. The paper has the potential to change the way many optimization problems are solved, and is certainly of interest to the NIPS community.

The first sentence of the abstract:
"It is well known that the optimal convergence rate for stochastic optimization of smooth functions is O(1/sqrt(T))".
(This is also repeated elsewhere, with a reference to Nemirovski and Yudin's book.)

As far as I am aware, this bound has never been shown. The lower bound has been shown for Lipschitz-continuous functions, but I'm not aware of such a lower bound on smooth functions so please provide a citation. Indeed, the recent work of Bach and Moulines ["Non-strongly-convex smooth stochastic approximation with convergence rate O(1/n)"] gives an algorithm for special cases that achieves an O(1/T) rate, indicating that this lower bound may not be true.

Note that an O(1/T) rate can also be achieved in some special cases, such as the one given above or cases where the noise goes to zero as the minimizer is approached (as with methods that use a growing sample-size).

"A small L2 regularizer does NOT improve the convergence rate of stochastic optimization."

Since this paper focuses on the case of smooth optimization, wouldn't adding a small L2 regularizer allow one to use the methods of [18,20] to achieve a faster (linear) convergence rate?

The first bullet point in in the introduction should indicate that the gradient is available too.

Please add to the paper that Definition 1 is equivalent to f' being beta-Lipschitz continuous.

It is worth adding a sentence to the paper pointing out that reducing the amount of regularization over time is closely-related to the classic proximal-point algorithm.

In the conclusion, please comment on whether O(1/T^2) rates could be achieved by an accelerated method in the mixed optimization scenario, and whether linear convergence rates could be achieved in the strongly-convex case.

It is worthwhile to add a sentence to the paper contrasting the mixed optimization approach with the very closely-related approaches that use growing sample-sizes of Friedlander and Schmidt ["Hybrid Deterministic-Stochastic Methods for Data Fitting"] and Byrd et al. ["Sample size selection in optimization methods for machine learning"].
Summary: I found this to be an interesting contribution to the literature on optimization for machine learning. The authors consider a new optimization framework that appears to practically useful, and provide a new algorithm that improves on state of the art convergence results. It could have been stronger if the method was shown to be competitive empirically with growing sample-size strategies as well as fast stochastic gradient methods like [18].

Submitted by Assigned_Reviewer_4

This paper considers the possibility of getting 1/T rates for optimization of a smooth function (which is itself an average of smooth functions), given access O(ln T) calls to a deterministic full-gradient oracle, and the rest of the queries to a stochastic oracle that returns one of the random loss functions (not just its gradient).

I think that the paper is put in context of other works very well. It is clear, of high quality, novel, original and definitely of significance (especially in settings where you are willing to pay for an expensive exact gradient once in a while, but not too often). However, more effort can be put into making the proofs more understandable (though many of the explanations are very good).

Some typos/questions:
Line 167 and 206: Are those \bar{w}_k rather than w_k?
Line 178: Should that be T_k + 1 rather than T+1?
Lines 216-229: Please use f for functions and g for gradients. Using g with different scripts for functions and gradients and other sums of functions or gradients is unnecessarily confusing.
Line 294: Do you mean using theorem 2?
Proof of Thm 1: Please specify which equations are being combined around the "hence", "since", "we obtain", etc for ease of following.
Line 316: epoches -> epochs
Summary: I recommend the paper for acceptance because I think it is well written, original, fairly easy to follow, and relevant to the NIPS audience.

Submitted by Assigned_Reviewer_5

DETAILED COMMENTS:

This theoretical work considers a setting of convex optimization where the algorithm can access both the stochastic and deterministic oracles. The authors show that with O(ln T) calls to the full gradient and O(T) calls to the stochastic oracle, the proposed mixed optimization algorithm can achieve an optimal O(1/T) rate of convergence for Lipschitz smooth functions.

In the introduction (L65-67), the comparisons of convergence rates between stochastic and deterministic methods are not fair. In the stochastic setting, the rate is typically expressed in the form of expected functions, while the faster rates of deterministic GD or accelerated variants are only for the empirical loss, i.e. the optimization error, and these faster rates do not necessarily lead to better generalization performance.

In Theorem 1, the rate is given on G(w). This function defined in (1) is the empirical loss, but not the expected function \int_{x,y} g(x,y) P(x,y) which is typically used in the guarantees of other stochastic algorithms. In other words, this rate is only for the optimization error, but not the generalization or excess error. It makes the contribution of O(1/T) questionable, since there exist stochastic algorithms minimizing smoothing functions that can achieve O(1/T^2) rate in the optimization error, see e.g. "Guanghui Lan, An optimal method for stochastic composite optimization (2012), in: Mathematical Programming, 133:1-2(365-397)".

Since MIXEDGRAD is pretty straightforward to implement, some experimental results are deserved to validate the theoretical analysis.

----------
Comments after rebuttal:
Yes, I'm aware of the deterministic objective that the authors focused on, and I believe that the O(1/T) optimization error bound for the optimization error is new. As I stated in my comments, I'm concerning the unfair comparisons (table 1) in the introduction, which might mislead some readers that are new to this field. The O(1/sqrt(T) bound for SGD or Lan's AC-SA is for the expected function, a direct measure of the generalization capability, which is not comparable with (1), even though the whole purpose of this paper is to reduce the optimization error. Moreover, when applying the trained model to testing data, the O(1/sqrt(t)) bound might still appear, due to the noise of the training data.
Summary: This theoretical work considers a setting of convex optimization where the algorithm can access both the stochastic and deterministic oracles. The proposed MIXEDGRAD algorithm is interesting, but experimental results are deserved. The theoretical guarantee of O(1/T) rate is only for the optimization error, and is slower than the O(1/T^2) rate that is achieved by other state-of-the-art algorithms.
Author Feedback

Author rebuttal: We are grateful to all anonymous reviewers for their useful and constructive comments and PC members for handling our paper.

========= Reviewer 1 ==========
Q: O(1/\sqrt{N}) for stochastic optimization of smooth function has never been shown

A: Indeed, it has been shown in Nemirovski and Yudin's book that using a first-order method, the effect of the stochastic noise cannot be decreased with a better rate than $O(1/sqrt{N})$ and this limitation is also valid when the function is smooth.
We agree with the reviewer about the special cases and we will definitely make the point more clear as you suggested. We also thank for the pointer to Bach and Moulines's paper.

Q: Adding few facts and missing references to the paper

A: Thanks for the great and insightful suggestions, facts and pointers to the missing references. We will consider all these comments.


========= Reviewer 2 ==========
We thank the reviewer for his comments. We will fix and apply them. Also we will do our best to making the proofs more understandable as suggested.


========= Reviewer 3 ==========
Q: The faster rates of deterministic GD or accelerated variants are only for the empirical loss, i.e. the optimization error, and these faster rates do not necessarily lead to better generalization performance.

A: Our study is focused on the optimization error, not the generalization performance or excess risk of the resulting prediction function. Optimization by itself is important to machine learning and has been studied recently by many theoretical and applied works such as Pegasos for stochastic SVM learning and Stochastic Average Gradient (SAG) [18] and Stochastic Dual Coordinate Ascent (SDCA) algorithms [20] to achieve better convergence rate.

We also note that good generalization can be achieved by combining efficient optimization with cross validation. Additionally, for most stochastic optimization strategies, although the bound is provided for expected loss function, the generalization error bound often involve unknown parameters (e.g. the size of the domain or the choice of step size) that also require tuning through cross validation.


Q: There exist stochastic algorithms minimizing smoothing functions that can achieve O(1/T^2) rate in the optimization error, see e.g. "Guanghui Lan, An optimal method for stochastic composite optimization (2012), in: Mathematical Programming, 133:1-2(365-397)".

A: We agree that few works such as Lan's Modified Mirror Descent method improves the bound for stochastic optimization, but still there is a dominating $\sigma/sqrt{T}$ term in the obtained bounds which is unavoidable in stochastic optimization of non-strongly convex objectives and which holds in our setting only when $\sigma=0$ which corresponds to having access to full gradient oracle for all $T$ iterations. In this case Lan's algorithm is equivalent to other optimal algorithms which achieve an $O(1/T^2)$ convergence rate when applied to smooth functions.

To show the difference between our algorithm and Lan's algorithm in terms of efficiency for stochastic optimization, let's count the number of calls to the full gradient oracle: to achieve an error of $\epsilon$, Lan's algorithm needs to make $1/\sqrt{\epsilon}$ calls to the full gradient oracle to achieve the desired optimization error, while with the help of cheap stochastic gradients, our algorithm only needs to call the full gradient oracle $\log(1/\epsilon)$ times and the stochastic oracle for $O(1/\epsilon)$ times. In other words, in order to get an $O(1/T^2)$ convergence rate by Modified Mirror Descent, it requires to access the full gradient oracle $O(T)$ times which is same as optimization in batch setting by Nesterov's optimal algorithm for smooth optimization. This point also raises the open question whether or not it would be possible to improve the convergence rate of the MixedGrad to O(1/T^2) by accelerated methods by calling the full gradient oracle only for $O(log T)$ times.

Q: Since MixedGrad is pretty straightforward to implement, some experimental results are deserved to validate the theoretical analysis.

A: We have preliminary experimental results on logistic regression which validates our theoretical findings and will be included in the extended version of this paper along with other extensions of the algorithm.